# The Relationship between Searches for COVID-19 Vaccines and Dynamics of Vaccinated People in Poland: An Infodemiological Study

**DOI:** 10.3390/ijerph192013275

**Published:** 2022-10-14

**Authors:** Anna Kłak, Konrad Furmańczyk, Paulina Maria Nowicka, Małgorzata Mańczak, Agnieszka Barańska, Urszula Religioni, Anna Siekierska, Martyna Ambroziak, Magdalena Chłopek

**Affiliations:** 1Department of Environmental Hazards Prevention, Allergology and Immunology, Medical University of Warsaw, Banacha 1a Street, 02-091 Warsaw, Poland; 2Institute of Information Technology, Warsaw University of Life Sciences, 02-776 Warsaw, Poland; 3Department of Gerontology, Public Health and Didactics, National Institute of Geriatrics, Rheumatology and Rehabilitation, Spartanska 1 Street, 02-637 Warsaw, Poland; 4Department of Medical Informatics and Statistics with e-Health Lab, Medical University of Lublin, K. Jaczewskiego 5 Street, 20-059 Lublin, Poland; 5Collegium of Business Administration, Warsaw School of Economics, 02-513 Warsaw, Poland; 6Department of Public Health, Institute of Psychiatry and Neurology, Sobieskiego 9 Street, 02-957 Warsaw, Poland; 7Graduate of the Faculty of Health Sciences, Medical University of Warsaw, Żwirki i Wigury 61 Street, 02-091 Warsaw, Poland

**Keywords:** COVID-19, epidemiology, infodemiology, SARS-CoV-2, vaccine, vaccinations, immunisation

## Abstract

Background: Google Trends has turned out to be an appropriate tool for evaluating correlations and prognostic modelling regarding infectious diseases. The possibility of selecting a vaccine against COVID-19 has increased social interest in particular vaccines. The objective of this study was to show dependencies between the frequency of searches for COVID-19 vaccinations and the number of vaccinated people in Poland, along with epidemiological data. Methods: Data were collected regarding Google searches for COVID-19 vaccines, the number of people in Poland vaccinated against COVID-19, the number of new cases, and the number of deaths due to COVID-19. Data were filtered from 27 December 2020 to 1 September 2021. Results: The number of new vaccinations smoothed per million correlated most strongly with searches for the word ‘Pfizer’ in Google Trends (Kendall’s tau = 0.46, *p* < 0.001). The number of new deaths correlated most strongly with the search phrase ‘AstraZeneca’ (Kendall’s tau = 0.46, *p* < 0.001). The number of new cases per million correlated most strongly with searches for ‘AstraZeneca’ (Kendall’s tau = 0.49, *p* < 0.001). The maximum daily number of searches ranged between 110 and 130. A significant interest in COVID-19 vaccines was observed from February to June 2021, i.e., in the period of a considerable increase in the number of new cases and new deaths due to COVID-19. Conclusions: A significant increase in interest in COVID-19 vaccines was observed from February to June 2021, i.e., in the period of gradually extended access to vaccinations, as well as a considerable increase in the number of new cases and new deaths due to COVID-19. The use of Google Trends with relevant keywords and a comparison with the course of the COVID-19 pandemic facilitates evaluation of the relationship between the frequency and types of searches for COVID-19 vaccines and epidemiological data.

## 1. Introduction

The first cases of the disease caused by the SARS-CoV-2 virus were reported to the World Health Organisation on 31 December 2019. Due to the rapid spread of the disease, WHO declared a pandemic of this disease caused by the virus (COVID-19) on 11 March 2020. The unknown aetiology of the disease, as well as unknown prevention and treatment methods, resulted in a variety of unsubstantiated reports emerging, leading to misinformation. The term “infodemic” then appeared, which describes the rapid spread of misinformation and fake news on the Internet [1]. Infodemiology has been defined in literature as “the science of distribution and determinants of information in an electronic medium, specifically the Internet, with the ultimate aim to inform public health and public policy” [2]. The objective of infodemiological studies is to describe how people search and navigate on the Internet to find health-related information, as well as how they communicate and share this information [3]. Infodemiology provides valuable insight into the health behaviours of the population in virtual reality [1,2,3]. In 2020, WHO developed a framework for managing the COVID-19 infodemic [4], which has been acknowledged by many public health organisations and WHO as an important emerging scientific field and critical area of practice during the COVID-19 pandemic.

The development of COVID-19 vaccines to protect against infection and a severe course of the disease became a challenge for scientists. The first vaccinations began in Poland on 27 December 2020 with health care professionals the first to be vaccinated with the Comirnaty vaccine (Pfizer/BioNTech) [5]. Access to vaccinations translated into a search for answers to various questions, e.g., are the vaccines safe? Can a person with autoimmune diseases vaccinate? What adverse events can occur?, etc. [6]. In Poland, the willingness to vaccinate against COVID-19 in the first period increased from 46% in November 2020 prior to the vaccination process, to 55% in March 2021 [7]. Next, older age groups were then offered vaccinations, with priority given to the oldest, teachers (AstraZeneca) and uniformed services [5].

The basic source of information on COVID-19 vaccinations in the period of reduced mobility due to the pandemic was the Internet, e.g., government websites publishing announcements about vaccinations, news websites, non-governmental organisation portals, as well as the websites of pro- and anti-vaxxers, including forums for exchanging information [6,8]. The possibility of selecting a vaccine increased interest in particular vaccines: mRNA or vector vaccine, one or two doses, and in the case of mRNA vaccines—Pfizer or Moderna [5]. The Polish government activities to promote vaccinations are also worth noting. From July to October 2021, every vaccinated person could take part in the “vaccine lottery” and win prizes [9]. The increased interest in vaccinations appeared with a new variant of the SARS-CoV-2 coronavirus. The basic question then was to what extent the previously administered vaccines would protect against new coronavirus mutations [10].

This paper presents the findings of the first infodemiological study carried out using Google Trends (GT) to evaluate dependencies between the dynamics of vaccinated people in Poland and the frequency of searches for COVID-19 vaccines. The study was conducted from 27 December 2020 (the onset of COVID-19 vaccinations) to the beginning of September 2021 (prior to the implementation of the booster dose). Google Trends has been used in scientific studies for several years. The first works that used this tool as the basic instrument in epidemiology were published in 2009, e.g., the evaluation of the spread of online information on bird flu compared to swine flu [11], the possibilities of the Internet to assess the development of the influenza epidemic [12], the usability of websites devoted to particular diseases [13]. Big data produced by Google Trends are an efficient source for analysing internet search behaviour, providing valuable insights into community dynamics and health-related problems. The analysis of Google Trends has turned out to be appropriate for evaluating the correlation and prognostic modelling within infectious diseases [14,15,16,17].

The findings of the study will reveal the behaviours of Polish Internet users in the context of searching for information on COVID-19 vaccines. Assuming that competencies within searching for health literacy information are nonuniform [18,19], and this is an essential skill during pandemics, these findings can serve as the base for the creation of strategies for managing public health information on the Internet and for the prevention of misinformation and fake news that impact unfavourable social behaviours. This study aimed at showing dependencies between the frequency of searches for COVID-19 vaccines and the number of vaccinated people in Poland. The specific objectives are:To show dependencies among the frequency of searches for COVID-19 vaccines and the number of vaccinated people in Poland;To show dependencies among the frequency of searches for COVID-19 vaccines and epidemiological data on SARS-CoV-2, i.e., the number of new cases and new deaths;To characterise the online interest in COVID-19 vaccines over a time basis.

## 2. Materials and Methods

### 2.1. Google Trends Data

This study was conducted in accordance with RECORD, the REporting of studies Conducted using Observational Routinely-collected health Data statement [20] and checklist (Appendix A). To examine the relationship between Google Trends and the number of new deaths, new cases and new vaccinations of COVID-19, data related to Google internet searches in Poland were downloaded from the Google Trends website (https://trends.google.pl/trends/?geo=PL, accessed on 5 September 2021), using the following keywords: ‘Moderna’, ‘Janssen’, ‘Johnson & Johnson’, ‘Novavax’, ‘Sputnik V’, ‘AstraZeneca’, ‘Pfizer’, ‘Comirnaty’, ‘BioNTech’, ‘mRNA’, ‘szczepionka mRNA’ (mRNA vaccines), ‘szczepionka przeciw COVID-19’ (a vaccine against COVID-19), ‘szczepionka na COVID-19’ (vaccine on COVID-19), ‘szczepionka COVID-19’ (COVID-19 vaccine), ‘rejestracja na szczepienie’ (registration for vaccination). For some keywords, no results were observed in Google Trends searches, and the results of the following terms were ultimately taken into account: ‘Moderna’, ‘AstraZeneca’, ‘Pfizer’, ‘mRNA’, ‘szczepionka mRNA’ (mRNA vaccines), ‘szczepionka na COVID-19’ (vaccine on COVID-19), ‘rejestracja na szczepienie’ (registration for vaccination). The keywords were selected based on their popularity on Google Trends during the study period. Data were filtered by location (Poland) for 249 days from 27 December 2020 to 1 September 2021. We chose this time span to cover the period from the first vaccine administration to the announcement of the Minister of Health of a booster dose (administered after the full vaccination schedule), even before the referral process for e-registration for the booster dose.

Google Trends data reflect a relative number of searches for a given term in relation to the total number of searches in Google. Numbers represent search interest relative to the highest point on the chart. A value of 100 is the peak popularity of the term (keyword), whilst a value of 50 means that the term has been half as popular. A value of ‘0’ indicates that insufficient data were collected for a given term, but it does not mean that users had not been searching for the given phrases. Searching for particular terms in Google Trends can be restricted to a country, region, place of residence and category.

### 2.2. Epidemiological Data of COVID-19

The corresponding data of COVID-19 on the daily number of new deaths, new deaths per million, new deaths smoothed per million, new cases per million, and new vaccinations smoothed per million during the same period were obtained from the GitHub database (https://github.com/owid/COVID-19-data/tree/master/public/data, accessed on 3 September 2021).

### 2.3. Statistical Analysis

The analysis was performed using the Kendall rank correlation coefficient (Kendall’s tau) to identify a monotonous character of dependencies between the pairs of variables. The asymptotic test was applied to evaluate the significance of the correlations. The significance level was stated at alpha = 0.05. In order to determine dependencies between the pairs of variables, Local Polynomial Regression Fitting was performed. The analyses were conducted using the R language.

## 3. Results

### 3.1. The Frequency of Searches for COVID-19 Vaccines vs. the Number of Vaccinated People

The number of new vaccinations smoothed per million showed the strongest correlation with the search for ‘Pfizer’ in Google Trends (Kendall’s tau = 0.46, *p* < 0.001, Figure 1), a strong positive correlation with ‘rejestracja na szczepienia’ (registration for vaccination) (Kendall’s tau = 0.34, *p* < 0.001) and ‘Moderna’ (Kendall’s tau = 0.31, *p* < 0.001), a weaker correlation with ‘AstraZeneca’ (Kendall’s tau = 0.18, *p* < 0.001), a considerably weaker correlation with ‘*mRNA*’ (Kendall’s tau = 0.13, *p* = 0.003) and ‘szczepionka mRNA’ (mRNA vaccines) (Kendall’s tau = 0.10, *p* = 0.03). There was no significant correlation with ‘szczepionka na COVID-19’ (vaccine on COVID-19) (Kendall’s tau = 0.01).

The Local Polynomial Regression model was adjusted to illustrate the strongest dependency between newly vaccinated people and the frequency of the search for the term ‘Pfizer’. A clear non-linear increasing dependency between these variables was observed (Figure 1).

### 3.2. The Frequency of Searches for COVID-19 Vaccinations vs. Epidemiological Data

The number of new deaths showed the strongest correlation with the search for ‘AstraZeneca’ in Google Trends (Kendall’s tau = 0.46, *p* < 0.001, Figure 2), a strong positive correlation with ‘Moderna’ (Kendall’s tau = 0.34, *p* < 0.001), a considerably weaker correlation with ‘rejestracja na szczepienia’ (registration for vaccination) (Kendall’s tau = 0.22, *p* < 0.001), a weaker correlation with ‘Pfizer’ (Kendall’s tau = 0.20, *p* < 0.001), ‘szczepionka na COVID’19’ (vaccine on COVID-19) (Kendall’s tau = 0.17, *p* < 0.001), ‘szczepionka mRNA’ (mRNA vaccines) (Kendall’s tau = 0.16, *p* < 0.001), and ‘mRNA’ (Kendall’s tau = 0.15, *p* = 0.001).

The Local Polynomial Regression model was adjusted to illustrate the strongest dependency between new deaths and the frequency of searches for the phrase ‘AstraZeneca’. A clear non-linear increasing dependency between these variables was observed (Figure 2).

The number of new cases per million showed the strongest correlation with the search for ‘AstraZeneca’ in Google Trends (Kendall’s tau = 0.49, *p* < 0.001, Figure 3), a strong positive correlation with ‘Moderna’ (Kendall’s tau = 0.29, *p* < 0.001), ‘szczepionka na COVID-19’ (vaccine on COVID-19) (Kendall’s tau = 0.16, *p* < 0.001), ‘rejestracja na szczepienie’ (registration for vaccination) (Kendall’s tau = 0.13, *p* = 0.004), ‘szczepionka mRNA’ (mRNA vaccines) (Kendall’s tau = 0.10, *p* = 0.003), and a statistically insignificant correlation with ‘Pfizer’ (Kendall’s tau = 0.07) and ‘mRNA’ (Kendall’s tau = 0.07).

The Local Polynomial Regression model was adjusted to illustrate the strongest dependency between new cases and the frequency of searches for the term ‘AstraZeneca’. A clear non-linear increasing dependency between these variables was observed (Figure 3).

The number of new deaths per million showed the strongest correlation with the search for ‘AstraZeneca’ in Google Trends (Kendall’s tau = 0.51, *p* < 0.001, Figure 4), a strong positive correlation with ‘Moderna’ (Kendall’s tau = 0.41, *p* < 0.001), a considerably weaker correlation with ‘rejestracja na szczepienia’ (registration for vaccination) (Kendall’s tau = 0.33, *p* < 0.001), ‘Pfizer’ (Kendall’s tau = 0.27, *p* < 0.001), ‘szczepionka mRNA’ (mRNA vaccines) (Kendall’s tau = 0.22, *p* < 0.001), ‘mRNA’ (Kendall’s tau = 0.20, *p* < 0.001), and ‘szczepionka na COVID’19’ (vaccine on COVID-19) (Kendall’s tau = 0.12, *p* = 0.006).

The Local Polynomial Regression model was adjusted to illustrate the strongest dependency between new deaths and the frequency of searches for the term ‘AstraZeneca’. A clear non-linear increasing dependency between these variables was observed (Figure 4).

### 3.3. The Characteristics of Online Interest in COVID-19 Vaccinations over a Time Basis

Figure 5 presents the searches for vaccinations from 27 December 2020 to 1 September 2021. The horizontal axis depicts the data from 249 days of the study. The time intervals of the frequency of the searches are very similar; the maximum frequency ranges from day 110 to day 130 of the study period.

### 3.4. The Characteristics of Epidemiological Data over a Time Basis

The maxima of epidemiological data (related to new cases and new deaths) appear from day 50 to day 150 of the study period and largely overlap with the maxima of the frequency of searches for vaccines. The maximum of the vaccinated people appears from day 150 to day 200 of the study period. It has been observed that the series of new deaths and new deaths per million are shifted to the right in relation to new cases per million, which means that the recorded deaths concern the cases from the earlier period. Detailed data are presented in Figure 6.

## 4. Discussion

The number of Internet users is growing worldwide. In 2021, there were 31.97 million Internet users in Poland, which accounted for 84.5% of the population [21]. The frequency of searching for health information online increases with the increasing number of Internet users [22]. The leading search engine in Poland is Google (chosen by 96.2% of users) [23]. External circumstances and the narrative in the media strongly influence behaviours of people looking for information on the Internet [24,25]. Restrictions limiting direct relationships during the pandemic, long periods of time spent at home and restricted access to general practitioners (GPs) have all intensified the search for health issues online [26,27].

This study has analysed the frequency of Internet searches for certain words and phrases in Google in the specific time interval in Poland. The first day of the analysis was the onset of COVID-19 vaccinations in Poland (27 December 2020) and the last day was 1 September 2021, when the Minister of Health announced a booster dose (administered after the full vaccination schedule), even before the referral process for e-registration for the booster dose. This study revealed dependencies between the frequency of searches for COVID-19 vaccinations and the number of vaccinated people. The BioNTech/Pfizer vaccine was the first to be approved for use in Poland (27 December 2020). By the week that the first batch of Moderna COVID-19 vaccines arrived in Poland (11–17 January 2021), there had already been 36 times as many BioNTech/Pfizer vaccines available [28,29]. The first peak of interest in BioNTech/Pfizer vaccines was observed with the start of the vaccination campaign, whilst in the case of Moderna vaccines, shortly after the start of this campaign. Further peaks were recorded between day 100 and day 180 of the study period. The Oxford-AstraZeneca COVID-19 vaccines (AstraZeneca) have been delivered to Poland on 6 February 2021 and introduced into use 6 days later. In the case of AstraZeneca, a significant increase was noted on day 75 (12 March 2021), i.e., at the time when some countries paused using this vaccine, which received a lot of publicity [30,31,32]. The order of the implementation of particular vaccines and their available quantity could impact their popularity and the result, showing that the number of new vaccinations smoothed per million correlated most strongly with the search of the word ‘Pfizer’ in Google Trends. On about day 100 (April 2021) the second significant peak of interest in the word ‘AstraZeneca’ was recorded, which coincided with the interest in the word ‘Pfizer’. It was also the peak of interest in the phrase ‘rejestracja na szczepienie’ (registration for vaccination). On 6 April 2021, people aged 40–59 years, who had sent the form to request the COVID-19 vaccine, received a referral for vaccination and the possibility to do it earlier [33]. Moreover, in mid-April, the Janssen COVID-19 vaccine (Johnson & Johnson) was introduced. The analysis showed a significant increase in the interest in COVID-19 vaccinations between February and June 2021, when apart from gradually improving access to vaccinations, a considerable growth of new cases and new deaths due to COVID-19 was noted.

During the COVID-19 pandemic, Google Trends was used to evaluate the behaviours of Internet users in many countries (e.g., India, Italy, the USA, Columbia, Turkey) [34,35,36,37,38,39,40]. In one part of the study the analysed period was the SARS-CoV-2 pandemic, and in the other part the words and phrases searched for before and during the pandemic were compared. The latter method was used by Rovetta A. and Castaldo L. who presented the interest in the words searched for from 1 September 2018 to 13 September 2021 and linked the increased interest in that period to different events impacting the change in the behaviours of Internet users in Italy [36]. At the time of the start of the COVID-19 pandemic, the seasonality of influenza had prevailed in search engines, then to be dominated by pandemic-related words.

Studies on GT by An L. et al. [24] showed that interest in the COVID-19 vaccines had increased in the first quarter of 2021 from 10% of searches related to the pandemic at the beginning of this period to 50%. There appeared interest in particular vaccines and their approval for use, which is also the case in this study; after the possibility to vaccinate against COVID-19 in Poland, the frequency of the search for specific vaccines increased.

It is worth noting that a study by Paguio J.A. et al. [40] carried out in the first period of COVID-19 (from 30 December 2019 to 30 March 2020) using Google Trends, revealed greater interest in vaccinations against other infectious diseases influencing the respiratory system, i.e., pneumococcus and influenza. At the end of 2020, the interest in COVID-19 vaccinations was observed. Vaccines raised hopes for the end of the pandemic, and concerns about their effectiveness and adverse effects [40]. A similar study was conducted by Maugeri et al. [41] who showed the advantages of using Google Trends data to predict the number of people vaccinated against COVID-19 and to monitor sentiments towards vaccinations.

Based on Google Trends data, Awijen H. et al. [42] presented data on social behaviours in 194 countries between 1 December 2020 and 4 March 2021. Pullan S. et al. [43] used GT to evaluate hesitancy in COVID-19 vaccine uptake and anti-vaxxer movements during the pandemic. They compared the development of vaccines with the intensification of anti-vaxxer movements. The study comprised the period from December 2019 to July 2020, i.e., before COVID-19 vaccines were approved for use. It was estimated that the interest in vaccinations remained high, but it was not able to define if it was positive or negative. Using the Internet for providing public health information based on reliable studies was deemed to be important. GT can serve to evaluate rapidly changing interests and perceptions of Internet users [43].

All articles on COVID-19 vaccinations pay attention to concerns about health and socio-economic situation, the role of decision-makers and the medical environment in raising social awareness of new vaccines, and limitations related to the use of Google Trends. The behaviours of Internet and Google search engine users were evaluated, which does not reflect the views of the entire population. Anonymity of Internet users does not allow determining their demographic characteristics, e.g., age, gender, education, socio-economic situation.

The presented findings are part of the first infodemiological study carried out using Google Trends to evaluate dependencies between the dynamics of vaccinated people and the frequency of searches for COVID-19 vaccines in Poland. This is the first such comprehensive study in Poland that combines the analysis of epidemiological data with Google Trends data in the context of the COVID-19 pandemic. The study also presents the dynamic of epidemiological data in Poland in the study period. One known limitation to our study was the fact that we did not analyse the relationship between the SARS-CoV-2 variants and the epidemiological data. The authors are of the opinion that this is a new direction of studies that should be taken into account in future and in more detailed studies. The next limitation of this study is the fact that different terms/phrases/keywords related to COVID-19 vaccines can show different strengths of correlation with epidemiological data. Searched terms and behaviours of Google users can change as the pandemic scenario is evolving, which results in the need to continuously update the model. This one concerned namely the qualitative analysis of interest in an argument in searching for the argument ‘vaccination’ (szczepienie). The observed increase in interest in searching was unexpected and can be explained by the interest caused by a “new” vaccine. A routine analysis of Google Trends can provide information on attitudes towards COVID-19 vaccinations, reluctance to vaccinations, and intention to vaccinate. Google Trends do not provide data for all places and therefore it would be difficult to develop a suitable model at the local level. The COVID-19 vaccines (BioNTech/Pfizer, Moderna and Oxford-AstraZeneca) were available to the public in Poland sequentially. The impact of the time gap between vaccines availability on the epidemiological data has not been analysed. Another limitation is the fact that knowledge on vaccinations is nonuniform in society, and the words searched for by Internet users may not reflect a real issue an Internet user would like to explore. The limited level of health literacy in the population may influence the fact that Internet users do not always type what they are actually looking for into a search engine. Health literacy enables a better understanding of health-related notions, and thus can improve the ability of a person to take responsibility for their health, which is of great importance in disease prevention [44].

## 5. Conclusions

The findings of our analysis confirm dependencies between epidemiological data (the number of vaccinated people, the number of new cases and the number of new deaths due to COVID-19) and the frequency of the searches for COVID-19 vaccines. The phrase that shows the strongest correlation with epidemiological data is ‘AstraZeneca’.

The analysis shows a considerable increase in the interest in vaccinations against COVID-19 from February to June 2021, i.e., in the period of the implementation of vaccinations in Poland and the third wave of the epidemic. The epidemiological data indicate a significant increase in new cases and new deaths due to COVID-19.

The use of Google Trends with relevant keywords, and comparison with the course of the COVID-19 pandemic and the vaccination campaign in Poland, allow the evaluation of the relationship between the frequency and type of searches on COVID-19 vaccines, and the dynamics of vaccinations and the number of cases. The analysis of the findings shows that health information in the media impacts the frequency of the queries on health typed into search engines, which should be taken into consideration by initiators of pro-health campaigns, e.g., encouraging to vaccinate against infectious diseases.

## Figures and Tables

**Figure 1 ijerph-19-13275-f001:**
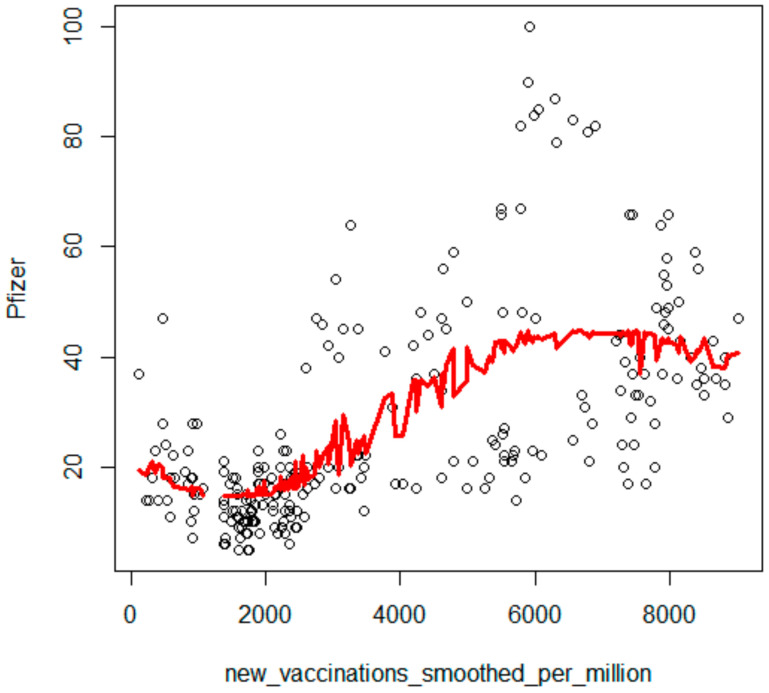
The frequency of searches for COVID-19 vaccinations vs. epidemiological data.

**Figure 2 ijerph-19-13275-f002:**
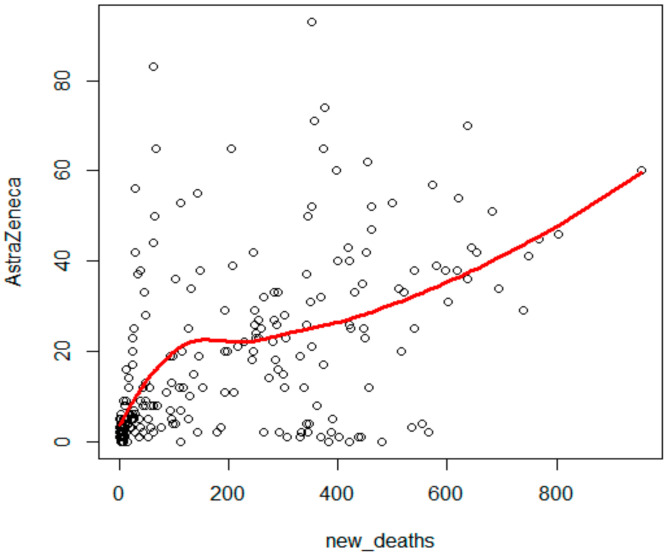
Local Polynomial Regression model of dependencies between new deaths and the search for the phrase ‘AstraZeneca’ in Google Trends.

**Figure 3 ijerph-19-13275-f003:**
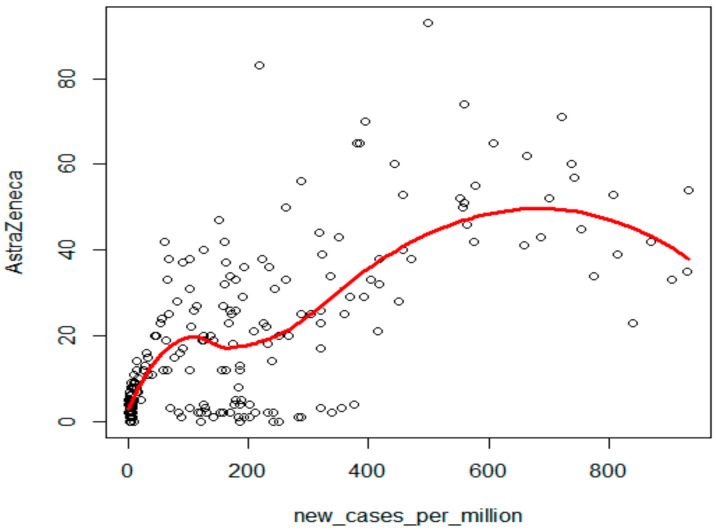
Local Polynomial Regression model of dependencies between new cases and the search for the term ‘AstraZeneca’ in Google Trends.

**Figure 4 ijerph-19-13275-f004:**
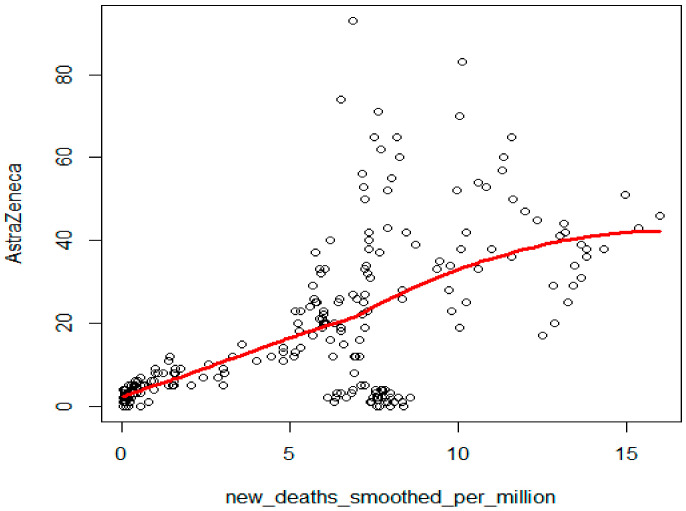
Local Polynomial Regression model of dependencies between new deaths and the search for the term ‘AstraZeneca’ in Google Trends.

**Figure 5 ijerph-19-13275-f005:**
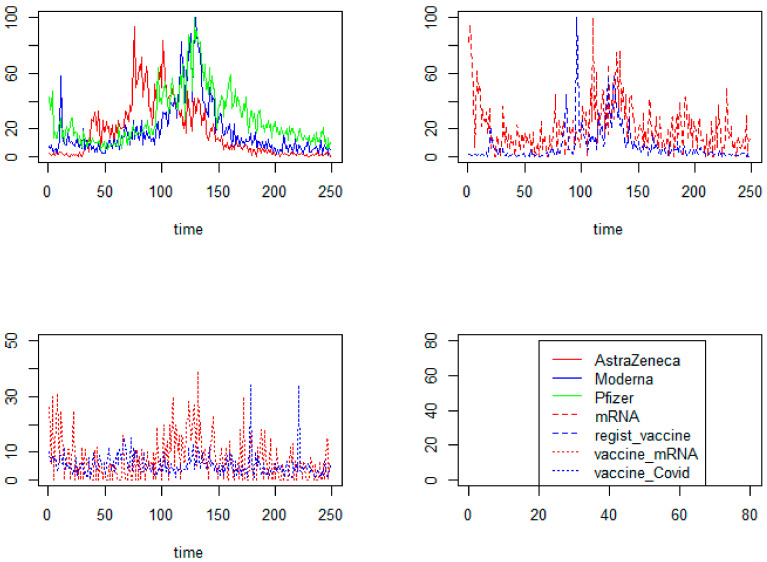
The frequency of searches for the terms ‘AstraZeneca’, ‘Moderna’, ‘Pfizer’, ‘mRNA’, ‘registration for vaccination’, ‘mRNA vaccines’, ‘vaccine on COVID-19’ in Google Trends in the period of 249 days (from 27 December 2020 to 1 September 2021).

**Figure 6 ijerph-19-13275-f006:**
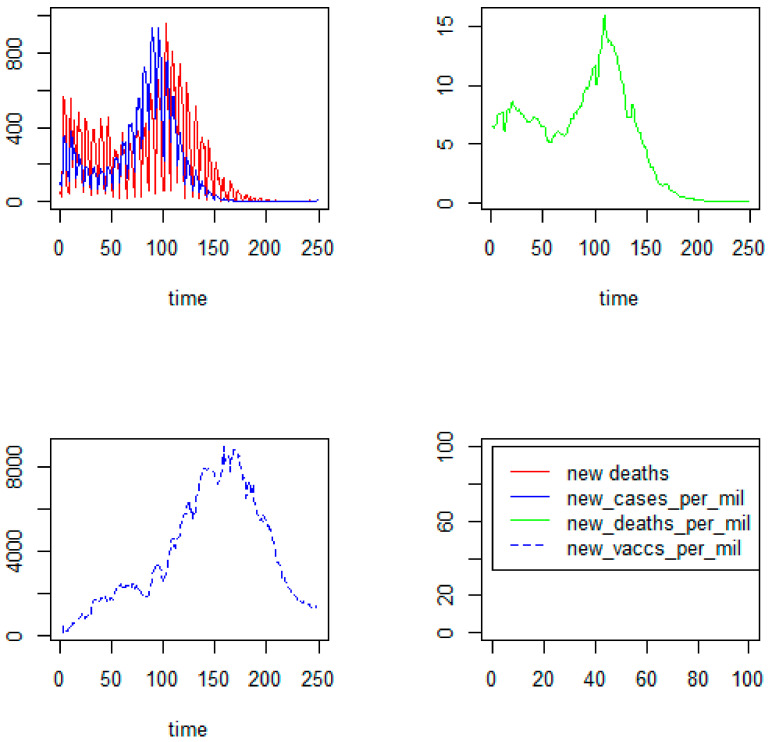
The characteristics of the COVID-19 epidemic in Poland in the period of 249 days (from 27 December 2020 to 1 September 2021).

## Data Availability

Data that support the findings of this study are available from the corresponding author, upon reasonable request.

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
