# Peer review of "The Relationship between Searches for COVID-19 Vaccines and Dynamics of Vaccinated People in Poland: An Infodemiological Study"

_ijerph, 2022, doi:10.3390/ijerph192013275_

Round 1

Reviewer 1 Report

Google Trends has been employed as an analytic tool in a number of recent studies that generate correlations and prognostic modelling to analyse the course of infectious diseases in human populations. In this retrospective study the authors sought to show the relationship between the frequency of searches for COVID-19 vaccines and the number of vaccinated people in Poland vaccinated against COVID-19, the number of new cases, and the number of deaths due to COVID-19. The authors provide a succinct introduction to the study by briefly outlining the history of this challenging pandemic and the use that people make of Internet searches to inform their choice of vaccine in a rapidly changing environment as vaccine availability changed.

The authors state that their study will be the first infodemiological study conducted employing Google Trends to evaluate dependencies between dynamics of vaccinated people in Poland and the frequency of searches for COVID-19 vaccines.

In Materials and Methods the authors, guided by Google Trends searching,  provide a comprehensive summary of their choice of keywords that relate closely to the vaccine names, vaccine types and pharmaceutical company names. The defined time span between 27 December 2020 and 1 September 2021 was chosen to cover a critical period between the first vaccine administered in Poland and the announcement of booster vaccine availability. These factors, relating to the filtering of extracted data, are critical to the understanding of the rationale of this particular study.

A sound description of the statistical testing of data is important and, as listed, this appears to be adequate for analysis of this type.

Results are presented in a logical order with adequate figures and accompanying informative legends. Some results are surprising and clearly depend upon the selected choice of words and filters applied.

The discussion covers the relevance of this study to the few other recent studies that have probed the behaviour of Internet users in several countries. The discussion provides a critical analysis of this study compared with any similar studies and most importantly the authors list and discuss the limitations of their study. Some of the limitations relate to the data actually available using Google Trends searches and these identified limitations would be of considerable assistance to other researchers planning similar investigations employing this useful tool.

Suggestions and clarifications

L22 an appropriate tool

L26 data were

L188 delete "'considerably"  since in both cases P<0.001

L191 a statistically insignificant

L187 and 191 uniformity of Kendall"s tau unless tau Kendalla is added to clarify a translation of Polish

L293 While this may be merely a personal choice I would find clearer if references were expressed as Awijen H. et al [42] showed that interest ....However this may be the guideline provided by the journal. I usually expect a date or reference number to appear after the author names.

L356 --> Author contributions? I suspect that these details will be filled in by the authors because at present the code does not relate to any identifiable author sequence.

Author Response

Dear  Reviewer,

Thank you very much for your reviews of our article “The relationship between searches for COVID-19 vaccines and dynamics of vaccinated people in Poland: an infodemiological study” (authors: KÅ‚ak Anna, FurmaÅ„czyk Konrad, Nowicka Paulina Maria, MaÅ„czak MaÅ‚gorzata, BaraÅ„ska Agnieszka, Religioni Urszula,  Siekierska Anna, Ambroziak Martyna, ChÅ‚opek Magdalena). We are grateful for your valuable suggestions and critical comments. We have exercised due diligence in order to refer to all the reviews. We have corrected the article according to your guidelines. Below is detailed information explaining how we have responded to the particular reviewers.

# Ad. Reviewer 1

We have corrected the article in accordance with all of your suggestions and clarifications.

We hope that we have referred to your reviews in a comprehensive manner and that the article has been thoroughly corrected. Thank you all once again for your valuable suggestions. We appreciate your time and look forward to your response.

All the authors approved the manuscript and this submission.

                                                                              Your sincerely,

                                                                           Anna KÅ‚ak, PhD

Reviewer 2 Report

The authors of this paper studied the relationship between searches for COVID-19 vaccines and the dynamics of vaccinated people in Poland. Generally, the paper is well written. The following comments should be considered when authors revise the manuscript.

1. The language quality of the paper needs to be polished further.

2. The motivations and highlights of the paper should be further strengthened.

Author Response

October 2, 2022

Reviewer of the International Journal of Environmental Research and Public Health

Dear Reviewer,

Thank you very much for your reviews of our article “The relationship between searches for COVID-19 vaccines and dynamics of vaccinated people in Poland: an infodemiological study” (authors: KÅ‚ak Anna, FurmaÅ„czyk Konrad, Nowicka Paulina Maria, MaÅ„czak MaÅ‚gorzata, BaraÅ„ska Agnieszka, Religioni Urszula,  Siekierska Anna, Ambroziak Martyna, ChÅ‚opek Magdalena). We are grateful for your valuable suggestions and critical comments. We have exercised due diligence in order to refer to all the reviews. We have corrected the article according to your guidelines. Below is detailed information explaining how we have responded to the particular reviewers.

# Ad. Reviewer 2

  1. The article has been subjected to proofreading.
  2. We have completed the discussion with the motivations, and highlights of the paper should be further strengthened.

We hope that we have referred to your reviews in a comprehensive manner and that the article has been thoroughly corrected. Thank you all once again for your valuable suggestions. We appreciate your time and look forward to your response.

All the authors approved the manuscript and this submission.

                                                                              Your sincerely,

                                                                             Anna KÅ‚ak, PhD

Reviewer 3 Report

The authors provided analysis and confirmed epidemiological data and vaccine-related searches on Google. There are several points that need to be addressed.

1. Are those three vaccines available to the public at the same time or sequentially? If they are not, can the time gap between availability lead to different dependencies with epidemiological data?

2. In addition to the available time point, are different vaccines equally available to the public from different regions within Polan? If so, Health care conditions in different regions may be another key factor for epidemiological data.

3. The epidemiological data may be primarily correlated to the emerging variants. Authors can include this part of the analysis and at least discuss it.

Author Response

October 2, 2022

Reviewer of the International Journal of Environmental Research and Public Health

Dear Reviewer,

Thank you very much for your reviews of our article “The relationship between searches for COVID-19 vaccines and dynamics of vaccinated people in Poland: an infodemiological study” (authors: KÅ‚ak Anna, FurmaÅ„czyk Konrad, Nowicka Paulina Maria, MaÅ„czak MaÅ‚gorzata, BaraÅ„ska Agnieszka, Religioni Urszula,  Siekierska Anna, Ambroziak Martyna, ChÅ‚opek Magdalena). We are grateful for your valuable suggestions and critical comments. We have exercised due diligence in order to refer to all the reviews. We have corrected the article according to your guidelines. Below is detailed information explaining how we have responded to the particular reviewers.

# Ad. Reviewer 3

  1. Thank you very much for this remark. Indeed, not all the starting dates of the use of individual vaccinations in Poland were mentioned. We have changed this. As in the majority of countries, COVID-19 vaccines were available sequentially. In the case of our analyses, we decided that analysing the exact dates of the availability of each vaccine was not significant. We were interested in the relationship between particular searches for the names of the vaccines with the number of vaccinated people, as well as in the relationship with epidemiological data.
  2. The data showing the exact spread of the availability of different types of vaccination over time is not publicly available, so we assume that the spread was even. In this study the data analysis was performed at the country level. As noted in the limitations of the study, Google Trends do not provide data for all places and therefore it would be difficult to develop a suitable model at the local level.
  3. The specific objectives of this study are:
  • to show dependencies between the frequency of searches for COVID-19 vaccines and the number of vaccinated people in Poland;
  • to show dependencies between the frequency of searches for COVID-19 vaccines and epidemiological data on SARS-CoV-2, i.e. the number of new cases and new deaths;
  • to characterise the online interest in COVID-19 vaccines over a time basis.

Investigating the relationships between the SARS-CoV-2 variants and the epidemiological data is not among the objectives of this study, however, it is an interesting issue worth being further explored. We believe this is an interesting new direction of studies that we can take into account in future analyses. We have not considered this issue in our paper, however we addressed this issue in the discussion.

We hope that we have referred to your reviews in a comprehensive manner and that the article has been thoroughly corrected. Thank you all once again for your valuable suggestions. We appreciate your time and look forward to your response.

All the authors approved the manuscript and this submission.

                                                                              Your sincerely,

                                                                               Anna KÅ‚ak, PhD

Round 2

Reviewer 3 Report

The revision and response answered my question. Thank you.